# Assessment of Mortality and Factors Affecting Outcome of Use of Paclitaxel-Coated Stents and Bare Metal Stents in Femoropopliteal PAD

**DOI:** 10.3390/jcm9072221

**Published:** 2020-07-13

**Authors:** Aleksander Falkowski, Hubert Bogacki, Marcin Szemitko

**Affiliations:** Department of Interventional Radiology, Pomeranian Medical University, 70-111 Szczecin, Poland; bakhis@hot.pl (A.F.); zrz@pum.edu.pl (H.B.)

**Keywords:** paclitaxel-coated stents, peripheral artery disease, restenosis, mortality

## Abstract

The use of drug-coated devices in intravascular therapy is aimed at preventing neointimal hyperplasia caused by excessive proliferation of vascular smooth muscle and thereby restenosis. Although its use seemed initially promising, a recent publication has shown an increased risk of mortality with paclitaxel-coated devices, and there is an urgent need to reaffirm assessments of drug-eluting stents (DES). Objective: The aim of the study was to compare mortality and effectiveness of paclitaxel-coated stents and bare-metal stents (BMS) in the treatment of peripheral arterial disease (PAD) with long-term follow-up. Materials and methods: In a single center randomized study, 256 patients with PAD were treated intravascularly with stent implantation. Patients were randomized into two groups: the first (n = 126) were treated with DES, and the second (n = 130) were treated with BMS. The study included evaluation after the procedure, after about 6 months and 36 months. Co-morbidities, with risks for atherosclerosis, were analyzed in all patients. Patients were evaluated for clinical outcome, restenosis frequency, and safety (complications and total mortality). Results: Clinical benefit at the end of the investigation was statistically significantly better in the DES group compared with the BMS group: 85.7% versus 66.2% (*p* = 0.0003), respectively. Restenosis occurred significantly less frequently in patients with DES: 16.0% versus BMS: 35.0%, *p* = 0.012. There was no significant effect of comorbidities on the frequency of restenoses. There were no differences in all-cause mortality over the three years with paclitaxel and no-paclitaxel stents cohorts (8.7% versus 7.1%; long-rank *p* = 0.575). No association was found with mortality and treatment with DES or BMS. Conclusions: The use of paclitaxel-coated stents gave good clinical benefit and caused a significantly lower frequency of restenosis compared to bare-metal stents. The use of paclitaxel-coated stents did not increase mortality.

## 1. Introduction

Peripheral arterial disease (PAD) is a global health problem associated with a 3 to 5-fold increase in mortality relative to the general population [1]. Severe claudication and critical limb ischemia (CLI) increase the risk of mortality, with mortality after five years in around 25% of symptomatic patients [2,3]. The main causes of death in patients with PAD are cardiovascular events and smoking-related diseases: cancers and respiratory diseases [3]. Endovascular treatment has become the primary treatment for PAD therapy.

Neointimal hyperplasia in a vessel that has been subjected to intravascular surgery is much faster than in cases of de novo lesions. For this reason, antimitotic drugs have been used with drug-coated balloons (DCB) and drug-eluting stents (DES), which reduce the frequency of restenosis caused by neointimal hyperplasia after surgery. The most commonly used drug has been paclitaxel, with its cytostatic action. When administered to the treated lesion, it interferes with cell growth and inhibits epithelial proliferation. The concept of preventing vascular endothelial hyperplasia (and thus restenosis) by delivering an antimitotic drug directly to the site of stenosis seemed to be a promising method, but recently, a sensational publication has been presented in which Prof. Katsanos showed increased mortality after treatment with drug-eluting devices [4]. There is therefore an urgent need to reaffirm previous assessments of paclitaxel-coated devices. The aim of the present study was to compare the safety and effectiveness of paclitaxel-coated and bare-metal stents in the treatment of PAD with long-term follow-up.

## 2. Materials and Methods

Patients (n = 256) who were to be treated for PAD were enrolled for the study. Patients who qualified for the procedure had symptoms of ischemic peripheral arterial disease diagnosed on the basis of clinical examination and confirmed by computed tomography angiography. Enrolled patients were classified between 2–5 on a Rutherford scale. Patients were recruited for the study from June 2016 to December 2016. Patients were randomized into two groups: the first (n = 126) were treated with DES and the second (n = 130) were treated with BMS. The study included evaluation immediately after the procedure, after 6 months and after an average of 36 months (±1 month) from the procedure.

The following inclusion and exclusion criteria were adopted:

Inclusion criteria: occlusion or narrowing (>70%) of the superficial femoral artery; lesion length between 40–200 mm; lesion only in the femoral-popliteal segment; at least one patent leg artery to the foot, without the need for simultaneous surgery in this area; no planned additional surgery over the next 30 days; no past amputation.

Exclusion criteria: presence of atherosclerotic lesions requiring surgery in the aorta, iliac or femoral arteries; presence of aortic, iliac or femoral vascular prosthesis; unusual anatomy or technical conditions that made placing the stent difficult; immunosuppressive therapy; allergy to acetylsalicylic acid, heparin or contrast agents; coagulation disorders; coronary artery surgery during the 30 days prior to inclusion or planned up to 30 days after inclusion.

### 2.1. Treatment Methods

Paclitaxel-coated stents (Zilver^®^ PTX^®^, Cook Medical) and standard, self-expanding, non-coated stents (Zilver^®^ FLEX, Cook Medical) were used for treatments.

Patients in both groups were recommended for anti-aggregation and anticoagulation therapy, both before, during and after the procedure. It was recommended to take acetylsalicylic acid and clopidogrel, each at a dose of 75 mg/day for at least 5 days before the planned surgery.

In patients already taking an oral anticoagulant (warfarin, acenocoumarol, dabigatran or rivaroxaban), the drug was replaced with low molecular weight heparin for a minimum of 5 days before surgery, and the previous dose was restarted after the procedure.

During the procedure, patients were routinely given 5000 IU of intra-arterial heparin. During the procedure, blood pressure was monitored, heart rate with a one lead EKG and oxygen saturation. Arteriography of the lower limbs was performed from contralateral puncture. A vascular introducer was installed (Cordis^®^, contralateral). Using the feature “road map”, the transition was made by means of a guide (Terumo Medical, stiff 0.035). A balloon catheter adapted to the diameter of the vessel and the length of the lesion was then introduced (Cordis^®^, Boston™, Medtronic™), and PTA was performed according to generally accepted principles. The stent was selected for the width of the vessel and the length of the stenosis in such a way that the diameter of the stent was 1 mm larger than the diameter of the treated vessel, and the stent covered the change over the entire length. Stents (40–140 mm long) were used with diameters 5–8 mm and delivery system 125 cm. After implantation, postdilation was performed with a balloon catheter. Follow-up arteriography was performed after the procedure. Hemostasis was maintained using an Angio-SEAL™ occlusion device (Terumo Medical Corporation, Tokyo, Japan) After the procedure, all patients were recommended to use two anti-agregational drugs: 75 mg/day acetylsalicylic acid indefinitely and 75 mg/day clopidogrel for 2 months.

### 2.2. Evaluation

All patients were examined 24 h before surgery, immediately after surgery, 6 months after surgery and on average after 36 months (±1 month) after surgery. Evaluations included clinical, hemodynamic and image analyses of vascular patency.

Chronic limb ischemia was determined according to the Rutherford scale. Successful stent implantation with possible residual postoperative stenosis of <30% was considered to be successful. A good clinical result was defined as an improvement of Rutherford classification by one category in comparison to the state before the procedure. Clinical benefit was defined as maintaining this result in the follow-up visit.

Major adverse events included death, amputation, clinically-driven target lesion revascularization, target-limb ischemia requiring surgical intervention or surgical vessel repair or the need for prolonged hospitalization. Mild complications were those that did not meet these criteria for serious complications.

Six months after the procedure and at the end of a patient’s observation, duplex ultrasonography and/or angio-computed tomography (CT) was performed. Restenosis was defined as >50% stenosis seen via duplex ultrasonography (with peak systolic velocity ratio <2.0) or from CT scan when available.

### 2.3. Safety Assessment

The basic safety assessment included a review of patients’ deaths for any reason or complications after the procedure. Causes of deaths were analyzed on the basis of medical documentation from the Clinics of the University Hospital or made available by external medical entities. In the absence of such possibilities, the cause of death was marked as undetermined.

### 2.4. Statistical Analyses

The following statistical methods were used: arithmetic mean and standard deviation; Czuprow’s convergence coefficient (Txy); statistical tests: Yates-corrected Chi-squared tests (for independent variables), U-Mann–Whitney tests and Kaplan–Meier survival analysis. Differences between the DES and BMS groups were analyzed for all patients and for patients with diabetes mellitus. Factors affecting the risk of restenosis were analyzed using single-factor and multi-factor Cox proportional hazard models. The relationship between the explanatory variables and the risk of restenosis was expressed using relative hazard ratios (HR) with a 95% confidence interval. The significance level was set at *p* = 0.05. All studies were conducted using commercial software (Statistica version 13.1; www.statsoft.pl).

## 3. Results

Patient characteristics showed only a significant difference between groups in terms of the length of the lesion being treated (Table 1).

Following arteriography immediately after the procedure, no case was found of early closure of the stent with treatment success in both groups estimated at 100%.

There were no serious complications after the surgery. However, mild complications were observed: these were small hematomas at the puncture site, had lacked significant clinical importance and did not require surgical intervention.

Clinical benefit after 6 months occurred in 232 patients (90.6%), specifically 121 patients (96%) in the DES group and 114 patients (87.7%) in the BMS group. Clinical benefit at the end of observation in both groups occurred in 195 patients (76.2%) and was statistically significantly more frequent in the DES group with 108 patients (85.7%) versus in the BMS group with 85 patients (66.2%); Χ^2^
*p* = 0.0003.

After 6 months, all patients underwent duplex ultrasonography. In the DES group, there were 3 cases of narrowing and 1 occlusion; in the BMS group, there were 6 cases of narrowing and 2 occlusions. After 36 months (±1 month), 64 patients with restenosis underwent duplex ultrasonography and 36 underwent CT. A total of 2 patients with restenosis only underwent CT. In the DES group there were 16 cases of narrowing and 4 occlusions; in the BMS group there were 37 cases of narrowing and 9 occlusions.

The frequency of restenosis at the end of observation was statistically significantly in patients from the BMS group (in the DES group in 20 patients, 16.0%; in the BMS group in 46 patients, 35.0%; Χ^2^
*p* = 0.012) (Table 2).

### 3.1. One-Way and Multivariate Effects on the Frequency of Restenoses in Each Group

Statistical analysis showed a lower number of restenosis during a 36-month follow-up after implantation of paclitaxel-coated stents. Restenosis was significantly more common in men than in women.

Assessment of the effects of hypertension, diabetes, a history of myocardial infarction, dyslipidemia, obstruction or stenosis and the patient’s age (on the day of surgery) on the frequency of restenosis at follow-up after 6 and 36 months showed no statistically significant effects of these factors (Table 3).

In multivariate analyses, taking into account factors significant in the univariate analysis (i.e., type of stent and gender), 36-month follow-up showed a statistically significant effect due to the type of stent (*p* = 0.03; HR = 0.2998 (0.556–2.16) but no significant gender effect was found (*p* = 0.066; HR = 0.451 (0.433–1.84)).

### 3.2. Test Results in Patients with Diabetes

The frequency of clinical patency and restenosis in patients with diabetes mellitus was analyzed in both groups (Table 4). In patients treated with DES, the results were significantly better at the end of follow-up with fewer restenosis in 3 patients in the DES group (6.5%) and with 14 patients in the BMS group (26.9%), Χ^2^
*p* = 0.0166; more clinical benefit was found in the DES group with 42 patients (91.3%) versus the BMS group with 35 patients (67.3%), *p* = 0.0082.

### 3.3. Survival Analysis

Twenty deaths were reported. In the DES group, 11 (8.7%) and in the BMS group, 9 (6.9%). According to sex: female/male mortality overall was 8 (8.5%)/12 (7.45%), in the DES group 4/7 and in the BMS group 4/5, respectively.

The probability of survival was high overall (Figure 1). The results obtained for mortality, BMS 8.7% vs. DES 6.9%, log-rank *p* = 0.5755, indicated no significant differences in the survival of patients treated with drug-releasing stents and non-drug-releasing stents (Figure 2). There was no relationship between the cause of death and treatment with DES or BMS (Txy dependency ratio = 0.00324) (Table 5).

## 4. Discussion

Initial research on drug-eluting stents (DES) showed significant reductions in lumen loss over 6 months compared with non-drug, bare-meal stents (BMS) [5]. Similar associations were demonstrated in studies with longer follow-up and large numbers of patients [6,7,8]. Additionally, “real-world” studies, such as a multi-center prospective, 5-year Japanese study, have shown good results with stents implanted with a drug: Freedom from target lesion revascularization (TLR) in the 60-month follow-up was 74.2% [9]. However, not all studies have shown such high frequencies with maintained patency, and some researchers have not shown significant benefits from drug use [10].

A comparative analysis using intravascular ultrasound after DES and BMS implantation in the femoral-popliteal region showed both lower intima hyperplasia and less vascular loss in the DES group, which may be evidence of higher efficiency of these stents in preventing restenosis [11,12]. In our study, the main endpoint was taken as restenosis and occurred significantly less frequently in the group of patients with implanted DES. By one-way analysis, it was shown that the type of implanted stent had a statistically significant effect on the frequency of restenosis (in favor of DES) during 36 months of observation, confirmed by a multivariate analysis.

In our study, the frequency of restenosis in the group with implanted BMS was 35%. This is an unsatisfactory value. Similar results have been presented by other researchers with restenosis frequencies of 19–37% [13,14,15]. Some studies suggest that the use of BMS has similar outcomes to using balloon angioplasty (PTA) alone [16].

Studies in large patient groups have confirmed a lower restenosis frequency when using paclitaxel stents, both in coronary arteries and in femoral popliteal sections [17]. In a five-year follow-up, Dake et al. found a >40% reduction in the risk of restenosis using paclitaxel-coated stents [18]. As in many works, our study has also shown that implanting stents with a drug gives better results than BMS.

In our study, the frequency of restenosis was not affected by other factors such as hypertension, diabetes, dyslipidemia, previous myocardial infarction, age on the day of surgery or multiple other factors for obstruction and stenosis. Previous research has shown very large discrepancies between conclusions from different studies on the effects of the above-mentioned risk factors on the frequency of restenosis. Results showing a higher frequency of restenosis in patients with diabetes have been previously reported [19]. Hiroyoshi Yokoi et al., in their study evaluating stents with paclitaxel, did not observe an association between the occurrence of pre-existing diabetes and the frequency of restenosis, which corresponds with the results presented in our study [9]. Similar research results have been presented in other publications [20,21]. Our study analyzed results in patients with diabetes in both stent groups: Significantly more frequent occurrence of restenosis was found in BMS-treated patients with diabetes during the 36-month follow-up in contrast to DES-treated patients. These conclusions are reflected in a meta-analysis conducted by Bangalore et al., which included patients with diabetes, where TLR was found to be much more common in cases with BMS than DES [22]. Similar results were observed in the work of Stettler et al. [23].

An important parameter evaluated during endovascular procedures is clinical benefit. A statistically significant difference was observed in the maintenance of clinical benefit in the group of patients with DES implants: At the end of the observation period, this was found in 85.7% in the group of patients with stents with paclitaxel, verses 66.2% in the BMS group. As in other multi-year studies comparing BMS to DES implants, in terms of clinical benefit, the stent with the drug was clearly better [18].

Most of these tests were performed prior to the announcement of the study showing higher total mortality after the use of paclitaxel devices [4]. The published meta-analysis from Katsanosa et al. aroused great concern. It showed an increase in mortality after the use of paclitaxel-coated devices compared to uncoated devices after 2 and 5 years of follow-up. There was therefore an urgent need to address the safety of the use of paclitaxel-coated devices. The US Food and Drug Administration has warned doctors to consider the risks and benefits of drug treatment and has given instructions for reporting any adverse events [24].

Our study did not show an increased total mortality in patients treated with DES and several previous analyses have also shown no association between paclitaxel-coated devices and higher mortality [25,26,27,28,29,30,31]. A multicenter, retrospective report from Medicare and Medicaid in the US showed no difference in overall mortality up to 600 days after treatment with and without paclitaxel-coated devices. However, these studies were based on data from an administrative database and were limited in data at the patient level. Clinical randomized trials at the patient level were recommended because the pooling of data from different studies has drawbacks. The US Food and Drug Administration has performed a meta-analysis of data from 5 years of three American randomized controlled trials (RCTs) of paclitaxel devices (IN.PACT SFA, Zilver PTX and LEVANT 2). After 5 years, mortality was 19.8% in patients treated with paclitaxel and 12.7% in patients treated with uncoated devices [32]. The FDA has acknowledged that this data may not be sufficient to characterize the presence, extent or causality of mortality. The FDA has stated that further observation is required, and there is still no clear evidence of a mechanism by which paclitaxel could cause mortality [33].

Paclitaxel is a commonly used cytotoxic agent used for intravenous administration as a chemotherapeutic agent in the treatment of malignant tumors at much higher doses than those used in PAD. Paclitaxel is a standard component of therapy in breast, ovarian, lung and cervical and pancreatic cancer. The safety profile of paclitaxel is acceptable even in the treatment of pregnant women during their second and third trimesters of pregnancy, without adverse effects on fetal development, and paclitaxel treatment for breast cancer does not increase cardiovascular mortality [34,35]. Compared to the above, a very low dose of paclitaxel is given in the treatment of PAD, and it is difficult to rationally explain a possible effect with increased mortality.

However, the authors of the meta-analysis argue that the half-life of the drug is important; it is 6 h when given intravenously but paclitaxel crystals on an endovascular device can have a half-life of weeks to months [4]. Given that thousands of cancer patients are receiving high dose paclitaxel and that many thousands of patients with coronary artery disease with drug stents are receiving it as well, it is hard to believe that mortality is higher in patients due to receiving a low dose exposure to paclitaxel in the lower extremities. Total doses of the drug in cancer treatments range from 236–392 mg [36], whereas the average levels of paclitaxel obtained by patients treated intravascularly is in the range of several milligrams.

In our study, the causes of death were various in both groups, and possible mechanisms that led to death from paclitaxel could not be found. However, further analysis is needed because DES treatment is already realized in widespread clinical practice.

### Limitations

A relatively small group of patients was analyzed. The patient population described in our study does not include all Rutherford classes or complex multi-level interventions. These more complex patients with increased comorbidities may have overwhelmed a mortality signal. The causes of death were not always evident. The studies were neither powered for mortality nor able to accurately assess potential links between concomitant medications and paclitaxel treatment. In addition, a small number of deaths limited the power of comparison.

## 5. Conclusions

The use of paclitaxel-coated stents gave good clinical benefit and caused a significantly lower frequency of restenosis compared to bare-metal stents. The use of paclitaxel-coated stents did not increase mortality. Further randomized studies are needed with large groups of patients and long-term follow-up.

## Figures and Tables

**Figure 1 jcm-09-02221-f001:**
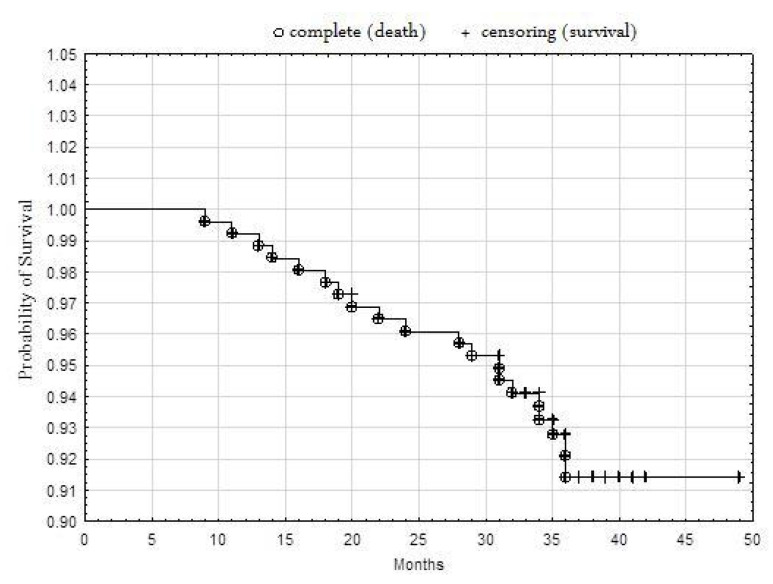
Kaplan–Meier survival analysis for all patients.

**Figure 2 jcm-09-02221-f002:**
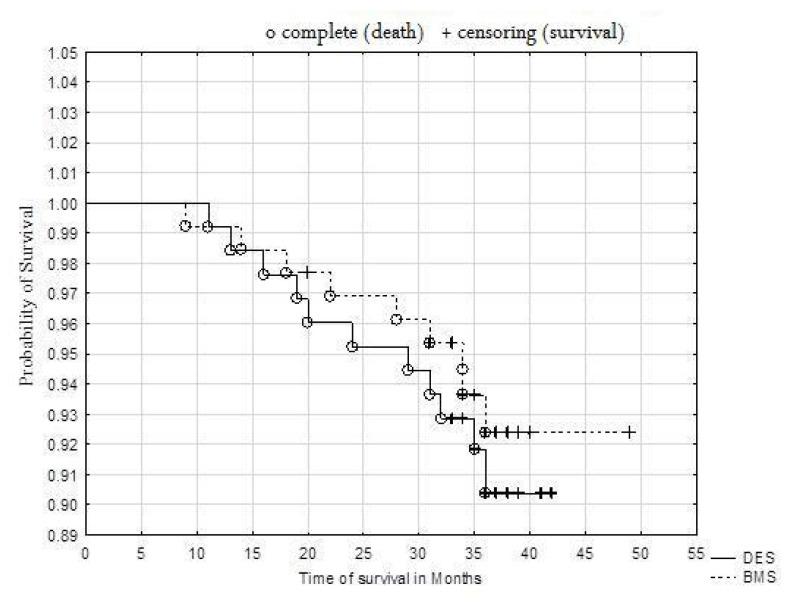
Kaplan–Meier survival analysis of patients treated with DES vs BMS. log-rank *p* = 0.5755.

**Table 1 jcm-09-02221-t001:** Baseline patient characteristics.

Patient Characteristics	DES Group	BMS Group	*p*-Value
Patients, n	126	130	
Age (Years)	66.6 ± 7.33	64.9 ± 8.10	*p* = 0.082
Sex (Male, Female)	80/46 (64%/36%)	82/48 (63%/37%)	*p* = 0.961
Rutherford Scale	2.82 ± 1.14	2.69 ± 1.03	*p* = 0.442
Obstruction/stenosis	50/76 (40%/60%)	62/68 (48%/52%)	*p* = 0.359
Hypertension	102 (81%)	106 (82%)	*p* = 0.932
Diabetes mellitus	46 (37%)	52 (40%)	*p* = 0.684
Coronary artery disease	56 (44%)	66 (51%)	*p* = 0.474
Myocardial infarction	28 (22%)	38 (29%)	*p* = 0.365
Stroke	14 (11%)	12 (9%)	*p* = 0.725
Dyslipidemia	74 (59%)	86 (66%)	*p* = 0.386
Chronic kidney disease	14 (11.1%)	16 (12.3%)	*p* = 0.814
Lesion length (mm)	93.8 ± 26.0	127.6 ± 49.7	*p* = 0.000

DES—drug-eluting stent; BMS—bare-metal stent; significance level *p* = 0.05.

**Table 2 jcm-09-02221-t002:** Comparison of the frequency of restenoses and the clinical benefit for both groups—drug-eluting stents (DES) and bare-metal stents (BMS).

Parameter	DES Group n = 126	BMS Group n = 130	*p*-Value
Clinical benefit			
after 6 months	121 (96%)	114 (87.6%)	*p* = 0.0276
after 36 months	108 (85.7%)	85 (65.4%)	*p* = 0.0003
Restenosis			
after 6 months	4 (3.2%)	8 (6.2%)	*p* = 0. 2596
after 36 months	20 (15.9%)	46 (35.4%)	*p* = 0.004
Target lesion revascularization			
after 6 months	4 (3.2%)	7 (5.4%)	*p* = 0.54
after 36 months	18 (14.3%)	41 (31.5%)	*p* = 0.001
Target limb amputation			
after 6 months	0	0	-
after 36 months	2 (1.6%)	4 (3.1%)	*p* = 0.684

Compared using Yate’s corrected Chi-squared; Significance: *p* < 0.05.

**Table 3 jcm-09-02221-t003:** Single predictor Cox analysis of the impact of many factors on the frequency of restenosis.

Factor	6 Months	36 Months
Type of stent	0.35 (0.04–3.34), *p* = 0.359	0.28 (0.10–0.84), *p* = 0.023
Sex	0.58 (0.08–4.10), *p* = 0.583	0.42 (0.18–0.98), *p* = 0.044
Hypertension	0.69 (0.07–6.59), *p* = 0.744	0.67 (0.25–1.82), *p* = 0.432
Diabetes mellitus	0.52 (0.05–5.01), *p* = 0.573	0.73 (0.30–1.79), *p* = 0.492
Myocardial infarction	0.94 (0.10–9.03), *p* = 0.956	0.83 (0.33–2.14), *p* = 0.705
Dyslipidemia	1.24 (0.43–3.64), *p* = 0.999	1.16 (0.47–2.86), *p* = 0.746
occlusion/stenosis	1.22 (0.17–8.63), *p* = 0.845	1.01 (0.42–2.39), *p* = 0.988
Age on the day of surgery	0.97 (0.86–1.10), *p* = 0.616	1.01 (0.42–2.39), *p* = 0.589

Cox proportional hazards model: (HR) with 95% confidence interval.

**Table 4 jcm-09-02221-t004:** Frequency of clinical benefit and restenosis in diabetic patients.

Parameter	DES Group	BMS Group	*p*-Value
	n = 46	n = 52	
Clinical benefit			
after 6 months	44 (95.7%)	45 (86.6%)	*p* = 0.227
after 36 months	41 (89.1%)	35 (67.3%)	*p* = 0.019
Restenosis			
after 6 months	1 (2.2%)	3 (5.8%)	*p* = 0.655
after 36 months	3 (6.5%)	14 (26.9%)	*p* = 0.017
Target lesion revascularization			
after 6 months	1 (2.2%)	3 (5.8%)	*p* = 0.62
after 36 months	2 (4.3%)	12 (23.1%)	*p* = 0.009
Target limb Amputation			
after 6 months	0	0	-
after 36 months	1 (2.2%)	3 (5.8%)	*p* = 0.62

Compared using Yate’s corrected Chi-squared; Significance: *p* < 0.05.

**Table 5 jcm-09-02221-t005:** Causes of death in both groups.

Cause of Death	Total	DES	BMS
Cardiovascular	9	5	4
Acute MI	2	1	1
Heart failure	4	2	2
Sudden cardiac death	1	1	0
Thrombosis	1	0	1
Stroke	1	1	0
Neoplasm	5	3	2
Non-malignant causes	4	2	2
Pulmonary	1	0	1
Renal	1	1	0
Infection	1	1	0
Trauma	1	0	1
Undetermined deaths	2	1	1
Total	20 (12.8%)	11 (8.7%)	9 (7.1%)

Coefficient of dependence Txy = 0.00324; no relationship between the cause of death and treatment.

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
