# Peer review of "Assessment of Mortality and Factors Affecting Outcome of Use of Paclitaxel-Coated Stents and Bare Metal Stents in Femoropopliteal PAD"

_jcm, 2020, doi:10.3390/jcm9072221_

Round 1

Reviewer 1 Report

Length of the lesion in relation to the outcome (or co-morbidity)?

Why this clinical difference after 6 months, in regard to what symptoms and what findings? Difference of restenosis at that time point was not sign. different?

DM has an impact on the outcome. Maybe other subgroups? Who else is at risk for restenosis? Do you think a tailored approach (DES only for patients at risk) is needed and justified, or DES always?

Author Response

Thank you for the critiques and suggestions.

1.Length of the lesion in relation to the outcome (or co-morbidity)?.

While the length of the lesion certainly impacts outcome, we believe that in this case it was not significant as the changes in both groups were in the same TASC classification type.

2.Why this clinical difference after 6 months, in regard to what symptoms and what findings? Difference of restenosis at that time point was not sign. different?

We also had the same question. Clinical benefit was defined as “maintenance of this result with freedom from persistent or worsening symptoms of ischemia (e.g. claudication, rest pain, ulcer, or tissue loss)”, which meant that the outcome was quite subjective and perhaps unclear. We decided to newly define clinical benefit in an objective manner – using the Rutherford classification. We analyzed our results again. In the BMS group there is still a greater loss of clinical benefit but not so significant. We suspect that this is because there are more patients with diabetes in this group. Because of this comorbidity, they have worse results, especially clinical, in part because of gradual deterioration of blood supply in arteries below the knees.

Fixed in manuscript.

3.DM has an impact on the outcome. Maybe other subgroups? Who else is at risk for restenosis? Do you think a tailored approach (DES only for patients at risk) is needed and justified, or DES always?

We only studied the outcome of stenting in DM because this is the most serious factor affecting results. Multivariate analyses did not confirm the effects of gender and other variables. However, I agree that creating separate comparative subgroups could give interesting results.

I think that if further research results show that the use of DES is absolutely safe, their use can be extended to everyone, because multiple analyses (including real-world)show better outcomes than BMS in all patients- in both those of higher and lower risk.

Reviewer 2 Report

To authors,

   In this randomized single center prospective clinical trial, the authors reported that paclitaxel-coated stent (Zilver PTX, Cook Medical) has a good clinical benefit and can achieve a significantly lower incidence of restenosis compared to bare metal stents in patients with peripheral artery disease (PAD). Also, the mortality during over 36-month follow-up was comparable between paclitaxel-coated stent and BMS treated group. The anti-restenotic effect of paclitaxel-coated device has been proved by number of clinical trials. However, as mentioned in this manuscript, the recent meta-analysis reported by Katsanos et al. demonstrated increased mortality after interventional procedure with paclitaxel-coated devices compared to uncoated devices in 2-5 years of follow-up. Within last two years, heated debates were conducted worldwide regarding safety of intervention with paclitaxel device. Although the mortality risk of paclitaxel device was refused in subsequent several clinical reports, the US FDA still recommends continuing diligent monitoring of patients who have been treated with paclitaxel devices. In this regards, the currently conducted unbiased head-to-head comparison for long-term mortality of the patients who received paclitaxel-coated stents and BMS is meaningful for clinicians. However, at the same time, this study contains multiple critical concerns to be improved as follows.

1.    (Page 1, Abstract); “endothelial hyperplasia” is not true. (Page 1, Introduction) “Proliferation of the endothelium” is not the mechanism of restenosis. These should be replaced with neointimal hyperplasia caused by excessive proliferation of vascular smooth muscle.

2.    The authors evaluated clinical benefit of intervention by improved symptoms. However, this evaluation is subjective and quite unclear. Readers cannot image what was the exact improvement of their symptoms, should be described in details. Quantitative assessment is also important such as ankle-brachial index (ABI) in pre- and post-treatment.

3.    The way for patient randomization should be written in Method section. When is the incorporating period of the patients in this study?

4.    The procedure of intervention (e.g. ballooning and stenting) should be described. Size and length of stent are quite important information. Should be added.

5.    Baseline patients’ characteristics (Table 1) should be moved from Materials and methods section to Result section. It seems unusual. Is the numbers in the table 1 mean or median value? Where are standard deviations or 25%-95% quartiles? Why lesion length was statistically different in two groups?

6.    The authors have to show exact data of CT and ultrasonography regarding restenosis. Only qualitative evaluation is not enough. Quantitative data should be added. The data of pre- and post- intervention is also indispensable. 

7.    (Figure 1); Y axis “Risk of restenosis” is confusing. It might be “Freedom from Restenosis”. Also, “Number at risk” should be changed in an appropriate term. Why the numbers at 40-month are 0?

8.    Why the authors abruptly showed the comparison of gender difference? It seems confusing. Also, in figure2, the males and females notation is switched.

9.    (Figure 4); What is the “censoring”? There is no information.

10. (Table5); What is the details of cardiovascular death (i.e. myocardial infarction, lethal arrhythmia, thrombosis, etc)? It should be described.

11. (Page 9, Limitation); I feel the limitation of this study is quite important section to be described. It should be improved.

12. The quality of English in the manuscript is not good enough. A lot of terms and the way of descriptions in this paper seemed not fitting in ordinal stent papers. For example, (Page 4, line 1-2); “statistically significantly more frequent”, (Page 8, line 9); “a multifactorial analysis”, (Page 7, line 10); ““real world”-type studies”, (Page 8, line 12-13); “There are even studies whose…….using balloon angioplasty (PTA) alone.”, (Page 8, line 25); “the occurrence of diabetes” It must be pre-existing diabetes. Most part of the main document should be carefully checked by the expertise researcher and totally re-written.

Author Response

Thank you for the critiques and suggestions.

1.(Page 1, Abstract); “endothelial hyperplasia” is not true. (Page 1, Introduction) “Proliferation of the endothelium” is not the mechanism of restenosis. These should be replaced with neointimal hyperplasia caused by excessive proliferation of vascular smooth muscle.

Fixed in manuscript.

2.The authors evaluated clinical benefit of intervention by improved symptoms. However, this evaluation is subjective and quite unclear. Readers cannot image what was the exact improvement of their symptoms, should be described in details. Quantitative assessment is also important such as ankle-brachial index (ABI) in pre- and post-treatment.

We agree that the outcome was too subjective. Clinical benefit was defined as “maintenance of this result with freedom from persistent or worsening symptoms of ischemia (e.g. claudication, rest pain, ulcer, or tissue loss)”, which meant that the outcome was quite subjective and perhaps unclear. We decided to newly define clinical benefit in an objective manner – using the Rutherford classification. “A good clinical result was defined as an improvement of Rutherford classification by one category in comparison to the state before the procedure. Clinical benefit was defined as maintaining this result in the follow-up visit.”

We analyzed our results again. This did not significantly change the results but became more clear for readers. 

This has been updated in the within the article.

ABI is an important study, but unfortunately we do not have complete ABI data pre- and post- treatment.

3.The way for patient randomization should be written in Method section. When is the incorporating period of the patients in this study?

Patients were recruited for the study from June 2016 to December 2016.

Fixed in manuscript

4.The procedure of intervention (e.g. ballooning and stenting) should be described. Size and length of stent are quite important information. Should be added.

Fixed in manuscript.

5.Baseline patients’ characteristics (Table 1) should be moved from Materials and methods section to Result section. It seems unusual. Is the numbers in the table 1 mean or median value? Where are standard deviations or 25%-95% quartiles? Why lesion length was statistically different in two groups?

Fixed in manuscript.

Numerical variables (age, Rutherford scale and lesion length) were expressed as the mean  ± standard deviations. Necessary corrections were added to the table.

This is the result we obtained. I think that this does not affect the results because patients in both groups were in the same TASC classification group type.

6.The authors have to show exact data of CT and ultrasonography regarding restenosis. Only qualitative evaluation is not enough. Quantitative data should be added. The data of pre- and post- intervention is also indispensable.

Fixed in manuscript

All patients were qualified based on CT. Occlusion / stenosis: DES 50/76; BMS 62/68 (tab.1)

7.(Figure 1); Y axis “Risk of restenosis” is confusing. It might be “Freedom from Restenosis”. Also, “Number at risk” should be changed in an appropriate term. Why the numbers at 40-month are 0?

8.. Why the authors abruptly showed the comparison of gender difference? It seems confusing. Also, in figure2, the males and females notation is switched.

We have made changes, but we agree that charts 1 and 2 are not necessary, because these data are presented in Table 3. For the sake of brevity, we decided to remove the charts

Fixed in manuscript.

9. (Figure 4); What is the “censoring”? There is no information.

Maximum likelihood estimation of the parameters of a normal distribution which is truncated at a known point. In this case survival.

Fixed in manuscript

10. (Table5); What is the details of cardiovascular death (i.e. myocardial infarction, lethal arrhythmia, thrombosis, etc)? It should be described.

Fixed in manuscript.

11. (Page 9, Limitation); I feel the limitation of this study is quite important section to be described. It should be improved.

Fixed in manuscript.

12. The quality of English in the manuscript is not good enough. A lot of terms and the way of descriptions in this paper seemed not fitting in ordinal stent papers. For example, (Page 4, line 1-2); “statistically significantly more frequent”, (Page 8, line 9); “a multifactorial analysis”, (Page 7, line 10); ““real world”-type studies”, (Page 8, line 12-13); “There are even studies whose…….using balloon angioplasty (PTA) alone.”, (Page 8, line 25); “the occurrence of diabetes” It must be pre-existing diabetes. Most part of the main document should be carefully checked by the expertise researcher and totally re-written.

Fixed in manuscript.

Reviewer 3 Report

Well done randomized trial with three year follow up which demonstrates the clinic benefit of DES compared with BMS without significant adverse events or increased mortality. Important addition to the literature given recent question of increased mortality with use of paclitaxel devices. 

Author Response

Thank you for the rewiews.

Round 2

Reviewer 1 Report

no comments

Author Response

Thank you for the review.

Reviewer 2 Report

To authors,

The authors have fixed previous manuscript in accordance with our criticisms. Most of the part seems to be improved significantly. However, there are still several issues.

  • (Table 1, 2, and 4); Some of the values were described as number (%) or mean ± SD, but others were not. Please keep consistency with the way of descriptions in number (%) or mean ± SD.
  • (Table 2 and 4); Other clinical evaluations such as target lesion revascularization, target vessel revascularization, and target limb amputation should be added on the table. Do you have any information about chronic kidney disease, the major risk for PAD?
  • (Table 3); Only describing p=value seems quite unusual. Add each hazard ratio and 95% confidence interval on the table. The result of multivariate analyses also should be put on.
  • (Figure 1 and 2); The term “completet” might be “complete”. P value should be added in the graph.
  • Writing was significantly improved, but there are still some typos, e.g. (Page 2, line 56); “obetween” should be “between”. Double periods were in the same sentence. (Page 2, line 86); “MedTronic” should be “Medtronic”. Please carefully recheck the whole part.

Author Response

(Table 1, 2, and 4); Some of the values were described as number (%) or mean ± SD, but others were not. Please keep consistency with the way of descriptions in number (%) or mean ± SD.

Fixed in manuscript.

(Table 2 and 4); Other clinical evaluations such as target lesion revascularization, target vessel revascularization, and target limb amputation should be added on the table.

fixed in manuscript, table 2 and 4 line 152, 174.

Do you have any information about chronic kidney disease, the major risk for PAD?

Fixed in manuscript table 1, line 130.

(Table 3); Only describing p=value seems quite unusual. Add each hazard ratio and 95% confidence interval on the table. The result of multivariate analyses also should be put on.

Fixed in manuscript table 3.

We calculated the multivariate analysis for two parameters and included this within the text.

(Figure 1 and 2); The term “completet” might be “complete”. P value should be added in the graph.

Writing was significantly improved, but there are still some typos, e.g. (Page 2, line 56); “obetween” should be “between”. Double periods were in the same sentence. (Page 2, line 86); “MedTronic” should be “Medtronic”. Please carefully recheck the whole part.

Fixed in manuscript.